# Knowledge, attitude, and practices of veterinarians towards canine vector-borne pathogens in Sri Lanka

**Ushani Atapattu**[1]*, **Vito Colella**[1], **Rebecca J. Traub**[1,2], **Anke Wiethoelter**[1]*

1 Melbourne Veterinary School, Faculty of Science, University of Melbourne, Victoria, Australia,
2 Department of Infectious Diseases and Public Health, City University of Hong Kong, Hong Kong SAR, China

* uatapattumud@student.unimelb.edu.au; anke.wiethoelter@unimelb.edu.au

## Abstract

Canine vector-borne pathogens (CVBP) have a worldwide distribution and show a high prevalence in tropical countries such as Sri Lanka. Some CVBP are zoonotic, with dogs identified as reservoir hosts for human subcutaneous dirofilariasis and potentially for spotted fever rickettsioses and re-emergent brugian filariasis in Sri Lanka, making these pathogens emerging public health issues in the country. Veterinarians are crucial in monitoring, preventing, and controlling these pathogens in dogs. Therefore, it is imperative to understand veterinarians' knowledge, attitude, and practices (KAP) regarding CVBP to mitigate their impact. A survey was designed and administered electronically to veterinarians residing and practising in Sri Lanka. Responses were evaluated using descriptive, univariable, and multivariable analyses to investigate associations between demographic factors, knowledge, attitude, and practices related to CVBP. Out of the 170 participating veterinarians, nearly 70% had moderate or high knowledge. However, the awareness of zoonotic pathogens, *Brugia* spp. (16%) and *Rickettsia conorii* (18%), was low, and a considerable number of veterinarians were unaware of the zoonotic nature of *Dirofilaria repens*. Based on multivariable analysis adjusting for experience and self-rated knowledge, new graduates had higher odds of knowledge compared to experienced veterinarians (OR 5.7, 95% CI 1.7–23, p = 0.028). Questions assessing the attitude towards CVBP indicated that most participating veterinarians comprehend and agree with their importance. Nearly all participants agreed that ectoparasite control is the best option to prevent CVBP infections (91%, 153/167) and that for effective treatment of CVBP, a definitive diagnosis is required (81%, 135/167). However, veterinarians recommended suboptimal treatments for some CVBP, like *Babesia gibsoni*. Better practices were associated with being a companion animal practitioner (OR 2.4, 95% CI 1.1–5.7, p = 0.032) and having a low to moderate canine caseload (OR 3.6, 95% CI 1.3–10.4, p = 0.038). Limited knowledge of zoonotic CVBP among veterinarians, along with suboptimal treatment, might contribute to dogs acting as reservoirs and high prevalence of these pathogens in Sri Lanka. Therefore, continued veterinary education is recommended to improve knowledge and practices, which in turn will help to improve the diagnosis, treatment, and control of these infections in Sri Lanka to ensure the well-being of dogs and humans.

**Data Availability Statement:** The authors confirm that all data underlying the findings are fully available without restriction. All relevant data are

within the paper and its Supporting Information files.

**Funding:** This work was supported by the University of Melbourne, Australia, through a Melbourne Research Scholarship to UA. The funders had no role in study design, data collection and analysis, decision to publish, or preparation of the manuscript.

**Competing interests:** The authors have declared that no competing interests exist.

## Author summary

Pathogens spread by ticks, fleas, and mosquitos, so called vector-borne pathogens, are common in tropical countries like Sri Lanka. Some vector-borne pathogens affect dogs as well as humans, potentially causing severe illness in both. As dogs can act as reservoir hosts, preventing these infections is vital for animal and human health. Veterinarians are the experts when it comes to diagnosing, treating, and controlling vector-borne pathogens in dogs. Thus, their understanding and approach to these diseases are key in reducing their impact. This study highlights both the strengths and gaps in veterinarians' knowledge, attitude and practices concerning vector-borne pathogens and identifies areas for improvement. This will assist relevant stakeholders and policymakers to develop targeted continuing veterinary education and best practice guidelines for diagnosis, treatment, and prevention of diseases caused by vector-borne pathogens in dogs. This will not only lead to better care for dogs affected by these diseases, but ultimately aid in reducing the risk to human health.

## Introduction

Vector-borne pathogens (VBP) infecting domestic dogs (*Canis lupus familiaris*) are widespread throughout tropical Asia [1]. These pathogens include bacterial species such as *Ehrlichia canis* and *Anaplasma platys*, as well as protozoan parasites such as *Babesia* spp. and *Hepatozoon canis* [1–4], with ticks being primarily responsible for their transmission. Dogs may have a role as reservoir for rickettsioses caused by several species of the genus *Rickettsia* [5], such as *Rickettsia conorii* [6] and *Rickettsia felis* [7]. Mosquitoes can also act as vectors of pathogens, such as filarial nematodes in the genera *Brugia* and *Dirofilaria*, which infect dogs and can pose a risk to humans [6,8–10].

In Sri Lanka, canine vector-borne pathogens (CVBP) are ubiquitous, with a high prevalence of *Babesia gibsoni* and *H. canis* [11] and a lesser but considerable prevalence of *A. platys*, *E. canis*, *Babesia vogeli*, and haemotropic mycoplasma species [11]. Furthermore, the high prevalence of zoonotic filariae in pet dogs such as *Dirofilaria* sp. 'hongkongensis' and *Brugia* sp. Sri Lanka genotype is contributing to an emerging public health issue [12,13], with Sri Lanka recently reporting the second-highest number of human subcutaneous dirofilariasis cases in the world [14]. Dogs in Sri Lanka have also been found to be seropositive for typhus and spotted fever rickettsioses (SFR) [15], which currently cause severe diseases in humans in Sri Lanka [16–19].

Sound knowledge and practices around CVBP are essential for veterinarians as they are involved in the diagnosis, treatment, and prevention of infections caused by these pathogens [20–23]. Currently, no information is available on veterinarians' knowledge, attitude and practices (KAP) around CVBP in Sri Lanka. This is a shortcoming in understanding what knowledge level is present, what recommendations are made, what practices are used, and whether those practices effectively mitigate the impact of these pathogens. A convenient and efficient method to assess this is KAP surveys [24], which have been used in health and health-related sectors. For example, several studies explored veterinarians' KAP related to zoonoses, canine vector-borne diseases, and One Health [22,25–30]. This study aimed to obtain an overview of KAP of veterinarians to provide a better understanding of potential enablers and barriers to the implementation of effective and efficient diagnosis, treatment, and control of CVBP in Sri Lanka.

## Materials and methods

### Ethics statement

This study was approved by the Office of Research Ethics and Integrity at the University of Melbourne (Reference Number 2021-20764-15614-3).

### Target population

The majority of veterinarians in Sri Lanka graduate from the Faculty of Veterinary Medicine and Animal Sciences at the University of Peradeniya (https://vet.pdn.ac.lk/), which is the only veterinary school in the country. To be able to work in Sri Lanka, veterinarians must meet the required qualifications according to the Veterinary Surgeons and Practitioners Act [31] and register with the Veterinary Council of Sri Lanka (VCSL). As of August 2020, 2041 veterinarians have been registered under the VCSL since the early 1950s, indicating the maximum possible size of the veterinary workforce in Sri Lanka [32].

### Survey design and data collection

Since veterinarians in Sri Lanka receive their education in English and are competent in the language, a KAP survey was designed and conducted in English using the Research Electronic Data Capture (REDCap) platform [33,34]. The survey was open to all veterinarians residing and registered to practice in veterinary or public health-related fields in Sri Lanka and is provided in **S1 File**. It comprised two main sections: the first section included core questions around demographic characteristics, knowledge, and attitude towards CVBP for all veterinarians. Demographic information collected included age, gender, graduation year, veterinary college, and primary discipline of practice. Data on attitude were collected on a 5-point Likert scale using statements around veterinary and public health implications and the importance of CVBP. The second part of the survey targeted canine practitioners, defined as veterinarians who either engaged exclusively in companion animal practice and treated dogs or any veterinarian who indicated that at least a quarter of their routine caseload consisted of dogs. Practices around diagnosis, treatment, and control of CVBP by these veterinarians were collected.

Seven volunteers tested the survey to ensure all questions were clear and easily understood prior to distributing the survey through social media (e.g., Facebook) and instant messaging platforms (e.g., WhatsApp and Viber). Informed written consent was obtained from each participant. Survey responses were collected from April to November 2021.

### Data analysis

Data were downloaded and cleaned using Microsoft Excel for Microsoft 365 MSO (Version 2212) and R version 4.2.0 in R studio Desktop (version 2023.3.1.446) [35] with 'dplyr' [36] and 'janitor' [37] packages. Responses less than 75% of the core questions were deemed incomplete and excluded from subsequent analysis. Explanatory variables were visualised and assessed using frequencies, bar charts and contingency tables utilising the packages 'janitor' [37], 'sjPlot' [38] and 'DescTools' [39]. Categorical variables were created for CVBP information sources ($\leq 3$, 4, $\geq 5$) and experience level based on graduation year ($\geq 2019$ –new graduate, 2014–2018 –moderately experienced, $\leq 2013$ –experienced). Responses were collapsed into three categories for the variables, canine caseload ($\leq 50\%$ –no-moderate, 51–75% –high, >75%–very high), confidence of Sri Lankan veterinarians handling CVBP (low, moderate, high) and self-rated knowledge on CVBP (low, moderate, high).

Using answers to 11 knowledge questions (**S1 File,** Questions 13–24), a knowledge score was calculated by allocating two points for each correct answer, one point for "don't know"

using it as a proxy for self-awareness of a knowledge gap, and zero points for incorrect answers. Knowledge of endemic VBP in Sri Lanka was assessed using a checklist (**S1 File**, Question 24), with one point allocated for each pathogen correctly identified as present or absent as of December 2021. The knowledge score was then categorised into "low" ($\leq 25$ points), "medium" (26–29), and "high" ($\geq 30$) and used as a response variable in subsequent uni- and multivariable analyses.

For canine practitioners, a practice score was calculated using seven questions around CVBP diagnosis (**S1 File**, Question 39), treatment (**S1 File**, Questions 40, 43–46), and prevention (**S1 File**, Question 42). Five-point Likert scale responses were collapsed with zero points allocated to "never" or "rarely", one point to "sometimes", and two points to "very often" or "always". Using the most effective drug for treatment based on peer-reviewed literature [40–42] was allocated two points, while other treatments were allocated zero points. The resulting CVBP practice score ranged from 0–14 points. It was categorised into "low-moderate" ($\leq 9$) and "high" ($> 9$) and used as a response variable for subsequent uni- and multivariable analyses.

Associations between demographic factors, attitude, and knowledge were investigated with ordinal regression using the 'MASS' package [43]. Logistic regression using the 'stats' package was applied to investigate associations between CVBP practices and attitude, knowledge and demographics. Likelihood P-values were obtained through the 'emmeans' package [44]. The variables, experience level, age, primary discipline, confidence in the profession, and knowledge scores were further collapsed as "new graduates or moderately experienced" and "experienced", "25–34 years" and "$\geq 35$ years", "companion animal practice" and "other disciplines", and "low-moderate" and "high", respectively to accommodate the limited number of participating canine practitioners.

A concept map, including ecological, socioeconomic and professional factors [45,46], was created to identify potential influences on the practices of Sri Lankan veterinarians around CVBP. To identify potential associations between explanatory variables such as demographics of the veterinarians and knowledge as well as practices around CVBP, directed acyclic graphs (DAGs) were constructed using DAGitty version 3.0 [47]. Those informed multivariable analyses.

Variables for multivariable models were selected based on univariable associations with at least marginal significance ($p \leq 0.25$) and were fitted using backwards-stepwise elimination. The independence of model terms was evaluated by calculating pairwise correlation coefficients with Cramer's V test using 'creditmodel' package [48]. Coefficients $\geq 0.6$ indicated high correlation, and only one of the variables was then included in the analysis. Confounding was investigated by adding variables individually back into the final model and assessing whether the model coefficients changed by $\geq 20\%$. This would suggest that the added variable substantially influences the relationship between the independent and response variables and thus acts as a confounder. For the adjusted ordinal regression model, adherence to the assumption of proportional odds was checked using the 'brant' package [49] and the model fit was evaluated using the Lipsitz goodness of fit test for ordinal responses [50] with the 'generalhoslem' package [51]. For the adjusted logistic regression model, the fitness of the model was validated using diagnostic plots (residuals versus fitted values, normal quantile-quantile, and residuals versus leverage). All illustrations used in this publication were constructed using Adobe Illustrator version 27.5 (Adobe Inc. USA).

## Results

Overall, 206 veterinarians responded to the survey invite, and 170 completed surveys were included in the analysis. Using data of veterinarians registered to practice in Sri Lanka [32], this relates to a response proportion of 8.3% (170/2041).

## Demographics of participants

Demographic details of the responding veterinarians (n = 170) are summarised in **Table 1.** Most veterinarians were females (58%, 99/170), 25–34 years old (75%, 128/170) and moderately experienced (59%, 101/170). All participants, except one, graduated from the University of Peradeniya. More than 70% (124/170) of the respondents were either involved in companion animal practice or were government veterinarians. While 84% (143/170) stated that they encounter dogs as patients, only 75% (128/170) either specifically focused on small animals or had a canine caseload of at least 25% and thus met our definition of canine practitioner.

## Knowledge on CVBP

The preferred method of Sri Lankan veterinarians to obtain information on CVBP was through textbooks (78%, 133/170) and colleagues (74%, 125/170), with 42% (72/170) using a maximum of three different types of information sources. Most veterinarians (>85%) were aware of the definition and concepts of CVBP (**S1 Table**). Nevertheless, only 52% responded correctly stating the vector of *Trypanosoma evansi* as flies, and 57% indicated correctly that *D. repens* infects humans (**S1 Table**). Over 70% of the veterinarians could correctly identify 10 of the 15 VBP listed as either present or absent in Sri Lanka (**S2 Table**). However, only 12%, 16%, 18%, and 32% were aware of the presence of haemotropic mycoplasmas, spotted fever *Rickettsia* (*R. conorii*), filariae of the genus *Brugia*, and *A. platys*, respectively. Overall, 33% (56/170) of Sri Lankan veterinarians had low CVBP knowledge scores, while 45% (76/170) had medium, and 22% (38/170) had high knowledge scores (**Table 1**). When asked to self-rate their knowledge on CVBP, 46% assessed their knowledge as low or moderate, while the remaining 54% regarded their knowledge as high. Comparing self-assessed and actual knowledge, 53% accurately gauged their knowledge level, while 29% overestimated and 17% underestimated their understanding of CVBP (**Table 1**).

## Attitude related to CVBP

**Fig 1** shows the responses of veterinarians to attitude-related statements. Most statements were met with high levels of agreement. Over 50% of veterinarians strongly agreed that good surveillance for CVBP (56%, 92/164) and vigilance regarding zoonotic CVBP (54%, 89/164) were important. Nearly all veterinarians (153/167) agreed that ectoparasite control is the best option to prevent CVBP infections, while over 80% (135/167) agreed that for effective treatment of CVBP, a definitive diagnosis is required. However, a greater proportion remained neutral (48%, 79/164) regarding their confidence in the profession to treat and manage CVBP infections in a similar fashion to high-income economies.

## Practices around CVBP infections by canine practitioners

The practices related to diagnosis, treatment, and prevention of CVBP by canine practitioners are described in **S3 Table**. Tick fever (babesiosis and ehrlichiosis) was the most frequently encountered CVBP infection, with 64% (81/126) of canine practitioners encountering daily cases, followed by filariasis (33%, 41/126). For trypanosomiasis, more than half of the respondents (52%, 65/126) reported that they had never diagnosed a case. Furthermore, the majority of veterinarians stated to experience good to excellent prognosis for tick fever (95%, 117/123) and filariasis (81%, 99/122). However, nearly half of the veterinarians were unaware of the prognosis of canine trypanosomiasis (49%, 60/123). While 66% (78/118) of canine practitioners utilised laboratories or other diagnostic facilities very often or always to diagnose CVBP, 43% (51/118) based their diagnosis on clinical signs. Monetary constraints when diagnosing

**Table 1. Demographic and knowledge details of Sri Lankan veterinarians participating in a survey on knowledge, attitudes, and practices around canine vector-borne pathogens (CVBP) (n = 170).**

| Variable | Category | n | % |
|---|---|---|---|
| Gender | Female | 99 | 58.2 |
| | Male | 67 | 39.4 |
| | Unknown | 4 | 2.4 |
| Age group (years) | 25–34 | 128 | 75.3 |
| | 35–44 | 25 | 14.7 |
| | 45–54 | 12 | 7.1 |
| | ≥ 55 | 5 | 2.9 |
| Veterinary college | University of Peradeniya | 169 | 99.4 |
| | Other† | 1 | 0.6 |
| Experience | New graduates (≥ 2019) | 16 | 9.4 |
| | Moderately experienced (2014–2018) | 101 | 59.4 |
| | Experienced (≤ 2013) | 53 | 31.2 |
| Primary discipline of practice | Companion animal practice | 66 | 38.8 |
| | Government | 58 | 34.1 |
| | Academia | 32 | 18.8 |
| | Other‡ | 14 | 8.2 |
| Species encountered | Dogs | 143 | 84.1 |
| | Cats | 131 | 77.1 |
| | Avian species (including poultry) | 87 | 51.2 |
| | Ruminants | 77 | 45.3 |
| | Wildlife species | 37 | 21.8 |
| | Swine | 35 | 20.6 |
| | Equine | 10 | 5.9 |
| | Rodents & rabbits | 4 | 2.4 |
| | None | 4 | 2.4 |
| | Fish | 2 | 1.2 |
| Canine caseload | No or low (0–24%) | 44 | 25.9 |
| | Moderate (25–50%) | 36 | 21.2 |
| | High (51–75%) | 49 | 28.8 |
| | Very High (>75%) | 41 | 24.1 |
| Canine practitioner | Yes | 128 | 75.3 |
| | No | 42 | 24.7 |
| Information sources used to update knowledge on CVBP | Textbooks | 133 | 78.2 |
| | Colleagues | 125 | 73.5 |
| | Other sources on the internet | 107 | 62.9 |
| | Publications | 103 | 60.6 |
| | Teachers/ academics | 80 | 47.1 |
| | Webinar | 50 | 29.4 |
| | Social media | 35 | 20.6 |
| | Workshops | 34 | 20 |
| | No sources | 3 | 1.8 |
| No. of information sources used | ≤ 3 | 72 | 42.4 |
| | 4 | 38 | 22.3 |
| | ≥ 5 | 60 | 35.3 |

(*Continued*)

**Table 1.** (Continued)

| Variable | Category | n | % |
|---|---|---|---|
| VBP knowledge score | Low | 56 | 32.9 |
| | Moderate | 76 | 44.7 |
| | High | 38 | 22.4 |
| Self-rated CVBP knowledge | Low | 6 | 3.5 |
| | Moderate | 73 | 42.9 |
| | High | 91 | 53.5 |
| Agreement between self-rated vs. actual knowledge | Underestimation | 29 | 17.1 |
| | Accurate estimation | 91 | 53.5 |
| | Overestimation | 50 | 29.4 |

[†]Madras Veterinary College

[‡]Includes farm animal practice, poultry, wildlife, and exotics

CVBP were encountered very often or always by 32% (38/118) of the participating canine practitioners. When treating canine babesiosis, most canine practitioners preferred imidocarb dipropionate (42% for *B. gibsoni*, 48% for *B. vogeli)*, while the preferred choice for canine ehrlichiosis was doxycycline (66%). For dirofilariasis, macrocyclic lactones (50%) and levamisole (41%) were indicated as the preferred choices.

Most canine practitioners (85%, 100/118) stated that they always inform dog owners about ectoparasite control, with topical fipronil (76%), systemic isoxazolines (63%), and propoxur powders (51%) as the products of choice to control tick, flea, and louse infestations. Overall, 54% (64/118) of canine practitioners had low to moderate CVBP practice scores, and 46% (54/118) had high CVBP practice scores (**S3 Table**).

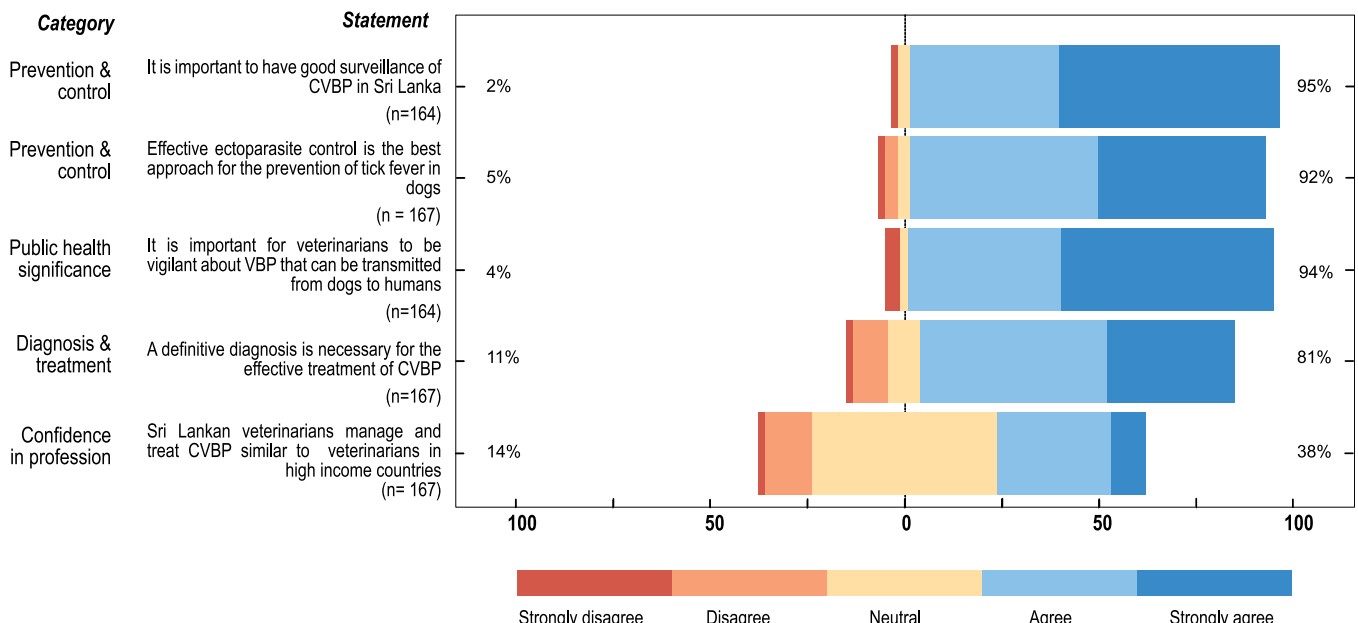

**Fig 1. Responses to statements evaluating attitude towards diagnosis, treatment, prevention, control, public health implications, and confidence in profession, regarding canine vector-borne pathogens (CVBP) of veterinarians in Sri Lanka.**

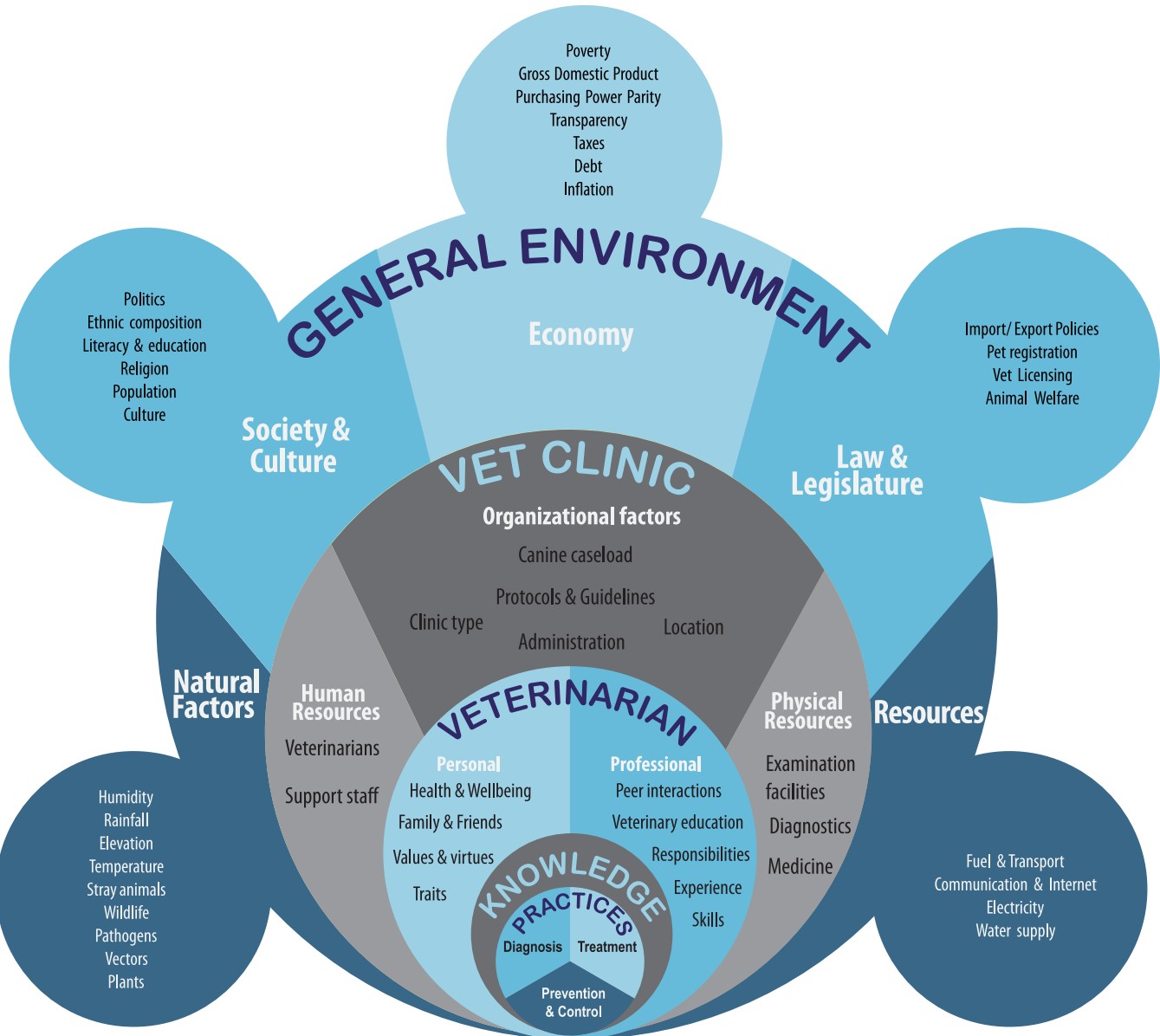

**Fig 2. Concept map identifying the interplay of ecological, socioeconomic and professional factors potentially influencing the knowledge, attitude, and practices of Sri Lankan veterinarians around canine vector-borne pathogens.**

### Factors associated with knowledge and practices around CVBP

A concept map identifying the complex interplay of macroenvironmental factors and those of the immediate environment that can directly or indirectly influence the veterinarians' knowledge, attitude, and practices related to CVBP is illustrated in **Fig 2**. Factors pertaining to nature, society and culture, economy, law and legislature, and resources were identified and reflect the general environment. Nested within the general environment are factors of the immediate environment, such as those associated with the clinic/hospital and the veterinarian themselves (**Fig 2**). While all these factors influence the diagnosis, treatment, and control of CVBP, not all of them can be changed in order to achieve improvements. Furthermore, not all influencers can be measured and quantified. Consequently, two DAGs depicted in **Fig 3A** and

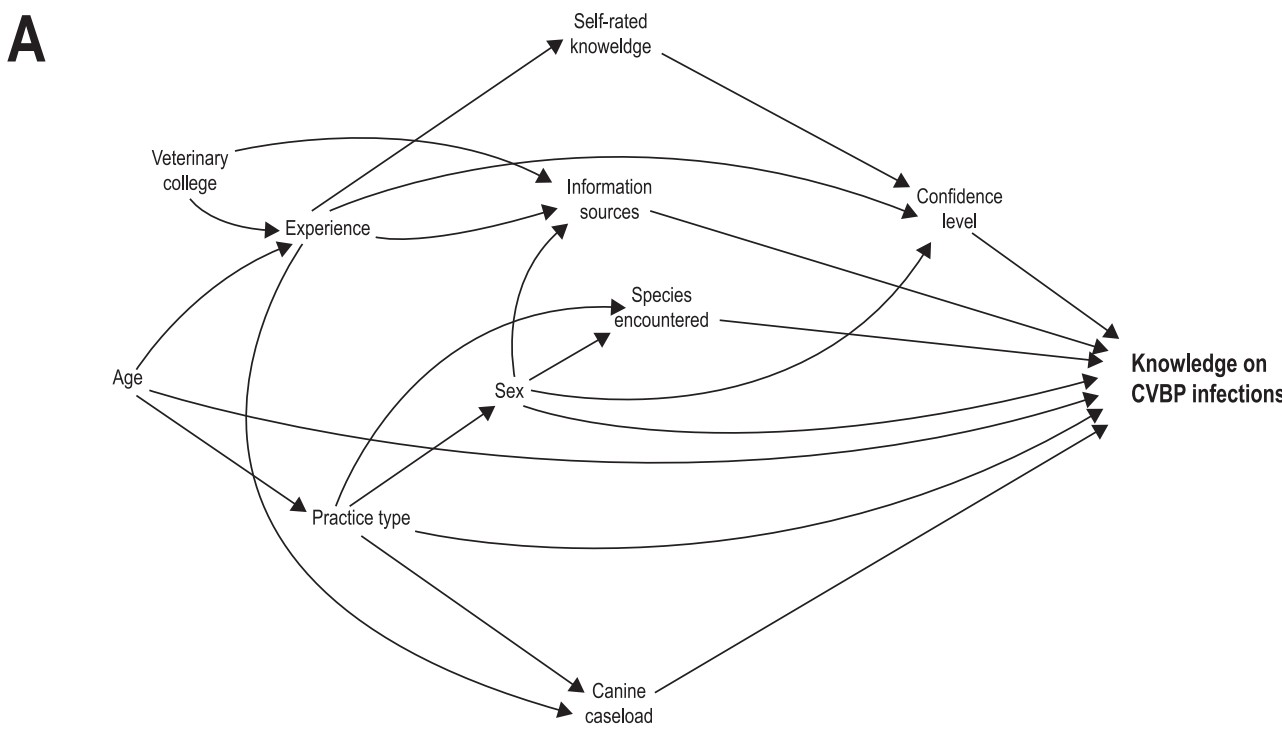

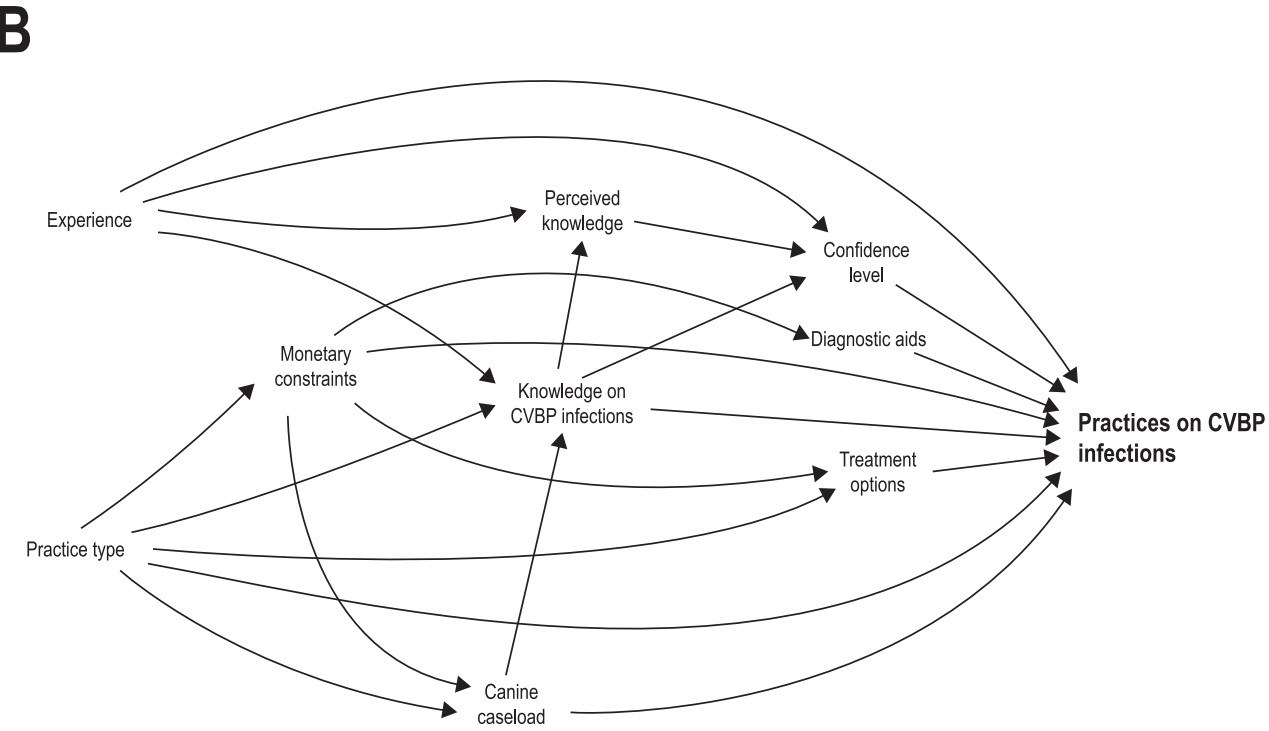

**Fig 3. Simplified directed acyclic graphs (DAG) depicting potential associations with A) knowledge of Sri Lankan veterinarians on canine vector-borne pathogens (CVBP) and B) practices of veterinarians in Sri Lanka related to CVBP infections.**

**Table 2. Final multivariable ordinal regression model for factors associated with knowledge on canine vector-borne pathogens of veterinarians in Sri Lanka (n = 166).**

| Variables | Estimate | SE | Odds Ratios (95% CI) | P-value |
|---|---|---|---|---|
| Intercept Low | -0.09 | 0.37 | | |
| Intercept High | 1.73 | 0.39 | | |
| **Gender** | | | | <0.001 |
| Female | 1.11 | 0.32 | 3 (1.6–5.7) | |
| Male | | | Reference | |
| **Experience** | | | | 0.028 |
| New graduate | 1.74 | 0.65 | 5.7 (1.7–23) | |
| Moderately experienced | 0.29 | 0.33 | 1.3 (0.7–2.5) | |
| Experienced | | | Reference | |
| **Self-rated knowledge** | | | | 0.028 |
| Low—moderate | | | Reference | |
| High | 0.68 | 0.31 | 2 (1.1–3.7) | |

CI = confidence interval; SE = standard error

Fig 3B were created, simplifying and delineating the associations between veterinarians' knowledge and the practices of canine practitioners related to CVBP, respectively.

In univariable analysis, gender, age, experience, primary discipline of practice, number of information sources used to update knowledge and self-rated knowledge were associated ($p \leq 0.25$) with the knowledge of CVBP (S4 Table). These variables were included in the initial multivariable ordinal regression model to determine factors associated with CVBP knowledge. As age and experience level of veterinarians were highly correlated, only experience level was included in the multivariable analysis. Table 2 shows factors associated with CVBP knowledge of veterinarians in Sri Lanka. Females had three times higher odds of knowledge compared to their male colleagues (OR 3, 95% CI 1.6–5.7, $p < 0.001$). Odds of knowledge in new graduates were six times higher (OR 5.7, 95% CI 1.7–23, $p = 0.028$) compared to experienced veterinarians. Those who rated their knowledge as high had higher odds of CVBP knowledge (OR 2, 95% CI 1.1–3.7, $p = 0.028$) (Table 2). When testing the final model, no confounders were identified as the coefficients of the final model from backward step-wise elimination did not change by $\leq 20\%$ when variables were reintroduced. The assumption of proportional odds was fulfilled ($p > 0.05$), and the Lipsitz goodness of fit test indicated a sufficient model fit ($p = 0.79$).

Univariable associations showed that practices around diagnosing, treating, and preventing CVBP were associated ($p \leq 0.25$) with the primary discipline of practice, the number of information sources used to update knowledge, and the canine caseload (S5 Table) and were included in the initial multivariable logistic regression model. The final multivariable logistic regression model evaluating practices of canine practitioners around CVBP in Sri Lanka indicated that veterinarians engaged exclusively in companion animal practice had higher odds of following best practices (OR 2.4, 95% CI 1.1–5.7, $p = 0.032$) than those involved in other practice types such as academia and government sector (Table 3). In addition, veterinarians presented with a low-moderate canine caseload had higher odds of having higher practice scores (OR 3.6, 95% CI 1.3–10.4, $p = 0.038$) compared to those who encountered very high canine caseloads (Table 3). No confounders were identified as the coefficients of the final model from backward stepwise elimination did not change by $\leq 20\%$ when variables were reintroduced. In addition, diagnostic plots indicated a good model fit.

**Table 3. Final multivariable logistic regression model for factors associated with the treatment, diagnosis, and prevention practices around canine vector-borne pathogen infections of canine practitioners in Sri Lanka (n = 118).**

| Variables | Estimate | SE | Odds Ratios (95% CI) | P-value |
|---|---|---|---|---|
| Intercept | -1.11 | 0.44 | | |
| **Primary discipline of practice** | | | | 0.032 |
| Companion animal practice | 0.89 | 0.41 | 2.4 (1.1–5.7) | |
| Other[†] | | | Reference | |
| **Canine caseload** | | | | 0.038 |
| Low—moderate (≤ 50%) | 1.28 | 0.52 | 3.6 (1.3–10.4) | |
| High (>50%—≤ 75%) | 0.26 | 0.46 | 1.3 (0.5–3.3) | |
| Very high (>75%) | | | Reference | |

[†]includes academia, government, wildlife, and exotics; CI = confidence interval; SE = standard error

## Discussion

This is the first comprehensive analysis of KAP related to CVBP of veterinarians in Sri Lanka. It provides valuable insights into how CVBP are diagnosed, treated, and prevented under the local context and highlights several avenues for continued veterinary education (CVE) to improve knowledge and practices. Such focused improvement can aid in uplifting veterinary services and formulating control protocols for CVBP, which are crucial to mitigate the veterinary and zoonotic impact of these pathogens.

In the present study, 67% of participating veterinarians had moderate or high knowledge scores. Compared to similar studies conducted with veterinary professionals in the states of Illinois and Ohio in the USA [22,25] and Mongolia [26], Sri Lankan veterinarians seem to be knowledgeable on CVBP. For instance, 75% of veterinarians in Illinois were only able to answer 10% of the knowledge questions on CVBP correctly [25], whereas, in this survey, 75% of the veterinarians in Sri Lanka answered 70% of the knowledge questions correctly. Only 43% of veterinarians in Ohio were able to correctly identify that *Ehrlichia* is present in their state [22], while the presence of *Ehrlichia* was correctly identified by 97% of Sri Lankan veterinarians. Furthermore, over half of the respondents in Mongolia had not heard of certain CVBP [26]. However, it is important to note that with their continental climate and cold winters [52], Illinois, Ohio and Mongolia have a lesser risk of CVBP burden compared to the tropical climate in Sri Lanka, making CVBP knowledge a vital component in the routine practice for Sri Lankan veterinarians.

Although over 75% of Sri Lankan veterinarians gave correct responses to nearly 70% of the knowledge questions in the survey, less than half responded correctly to questions on zoonotic VBP, demonstrating low awareness around this group of pathogens. Despite the notable incidence of human subcutaneous dirofilariasis caused by *D. repens* (recently genetically characterised [12] as *Dirofilaria* sp. 'hongkongensis') [53–59] and its high prevalence in dogs in Sri Lanka [12,60], only 57% of veterinarians identified this pathogen as a risk to humans. However, this proportion is higher compared to Baltic and Nordic countries, where only 34% of veterinarians considered this pathogen to be zoonotic [61]. Nevertheless, low awareness of *Brugia* species among Sri Lankan veterinarians is concerning as *Brugia* [62] in humans is re-emerging, and domestic dogs seem to act as reservoirs [12,60,63]. Similarly, dogs are known reservoirs of *R. conorii* in Sri Lanka [15]. Nevertheless, hardly 20% of Sri Lankan veterinarians were aware that *R. conorii* is endemic to Sri Lanka, which is considerably lower compared to veterinary students in Brazil (84%) [64]. Awareness of zoonotic VBP is not only crucial for veterinary professionals to minimise occupational health risks but also to mitigate their impact

on human and animal health [64]. Thus, a lack of knowledge among veterinarians could directly threaten the effective control of these pathogens [13,65,66].

The multivariable analysis identified females, new graduates, and those who self-rated their CVBP knowledge as high to have higher odds of knowledge. A comparable trend was evident from similar studies evaluating the knowledge, attitude and practices of VBP among veterinarians in Mongolia and the USA [22,26]. New graduates are likely to possess fresh and most up-to-date knowledge compared to older, more experienced veterinarians. Furthermore, Sri Lankan veterinarians appear to be accurate in their self-evaluation of knowledge. Similarly, veterinarians from Ohio, USA, also correctly rated their knowledge [28], possibly signifying that veterinarians in general have a good conscience about their level of knowledge.

The attitude-evaluating statements demonstrated that Sri Lankan veterinarians had a high agreement on the importance of preventing and controlling VBP through surveillance and ectoparasite control. A similar high agreement on the importance of zoonotic tick-borne diseases was also observed with veterinarians in Ohio, USA [28]. Such a positive attitude potentially supports veterinarians' willingness to mitigate the impact of these pathogens [22].

In addition to knowledge and attitude, we were able to provide an overview of practices related to CVBP diagnosis, treatment, and control in Sri Lanka. In concordance with the results of previous studies indicating a high prevalence of CVBP in Sri Lanka [11,12,60,67], veterinarians in this study reported frequently encountering CVBP cases such as tick fever and filariasis. Nearly half of the participating veterinarians stated that they would diagnose CVBP using only clinical signs without the aid of other diagnostics. A study of field veterinarians in Sri Lanka also indicated the infrequent use of diagnostic laboratories [68]. One possible explanation for the use of inadequate diagnostics could be monetary constraints encountered, as indicated by a considerable proportion of participants. In contrast, in Ohio, USA, over 70% of veterinarians employed external laboratory facilities to identify tick species discovered in their patients [22]. Not using adequate diagnostics can lead to incorrect diagnosis, as clinical signs of most CVBP are non-specific [69]. For example, inadequate diagnostic practices most likely caused less frequent diagnoses of *H. canis* despite its high prevalence [11]. Consequently, this could prevent the administration of appropriate treatment and the total clearance of the pathogen, thereby facilitating the emergence of reservoirs for CVBP. This is concerning as some of these pathogens are zoonotic, or potentially could infect animals of economic importance and vulnerable wildlife.

Evaluating treatment practices holistically, some of the CVBP were treated using substandard treatment protocols. For instance, most canine practitioners preferred imidocarb dipropionate to treat *B. vogeli* and *B. gibsoni*, even though its efficacy towards *B. gibsoni* is low to none [70]. While these substandard protocols may improve the patient clinically, their reduced efficacy may only lead to partial elimination of the pathogen, potentially creating reservoirs of infection and leading to relapses [71]. In addition, some treatments can cause adverse reactions [70]. For example, diminazene aceturate is effective against *B. gibsoni*, but its variable therapeutic index in dogs can lead to severe toxic manifestations [72], sometimes outweighing its therapeutic benefits. Most veterinarians recommended effective ectoparasiticides, with fipronil-based products being the preferred choice. However, when comparing the veterinarians' recommendations to the products administered by dog owners [11], they do not seem to comply with the recommendations made. A probable explanation might be the higher cost of the recommended products, as most dog owners mentioned cheaper but less effective options as their preferred choices [11].

Companion animal practitioners and those who reported to encounter a low to moderate proportion of canine cases had higher odds of better practices. They might be driven to portray better practices around CVBP compared to other veterinarians (e.g., academia and the

government sector) due to the changing role of dogs in Sri Lankan households over the past few decades [73]. Over 40% of dog owners now consider dogs as companions for themselves and their children [74], making them an integral part of their families. An increase in dog ownership and status invariably increases the demand for specialised veterinary services [73]. While one might expect increasing canine caseloads to lead to better practices, in this study, the best practices were observed by those who encountered low to moderate canine caseloads. One possible explanation for this might be that specialised companion animal veterinarians encounter not only dogs but also cats, pet birds, and pocket pets, thereby reducing their proportion of canine cases seen. On the other hand, government veterinarians performing anti-rabies vaccination and dog neutering [75] might have a very high proportion of canine cases.

Several limitations apply to this study. The total number of registered veterinarians [32] includes those who are out of practice, overseas, or deceased, likely decreasing our target population size and increasing the actual proportion responded. In addition, COVID-19 pandemic related events that took place during the surveying period potentially influenced the number of responses received. Nevertheless, the resulting response rate is low and subject to selection bias. In addition, having the survey exclusively online might have led to lower participation of less technology-savvy veterinarians. The high proportion of female participants could be explained by the feminisation of the veterinary workforce as demonstrated in several other studies [76–79]. Veterinarians having an interest in the topic might have been more likely to complete the questionnaire. Hence, participants likely represent an engaged, best-informed cohort regarding CVBP knowledge and practices in Sri Lanka. However, knowledge gaps and suboptimal practices were present among this cohort which highlights the necessity of CVE to develop good standards of knowledge and practices around CVBP in Sri Lanka. Another limitation of this study is that it is largely based on self-reported behaviour and therefore prone to measurement bias. Knowledge and practices around CVBP are subject to an intricate interplay of macroenvironmental influences, encompassing economic, ecological, legislative, and societal factors [80,81]. These factors are not only hard to measure but also challenging to alter. This study focused foremost on modifiable factors, even though they are not the sole contributors.

## Conclusions

Conclusively, veterinarians in Sri Lanka responded correctly to most knowledge questions on CVBP and have a positive attitude towards controlling and preventing these infections. However, their awareness of the zoonotic CVBP, drawbacks in diagnosis and treatment needs improvement. Therefore, we recommend raising awareness and providing opportunities for continuing education for veterinarians towards improving the veterinary diagnostic and treatment facilities, particularly in the government sector, to increase the quality and accessibility of veterinary services offered in relation to CVBP in Sri Lanka. Implementing cost-effective diagnostic protocols for effective CVBP diagnosis and drugs for treating and preventing these pathogens should be re-evaluated to apprehend deficiencies. Formulation of national standards for best practices on CVBP under varying local contexts would foster improved diagnosis, treatment, and control of these infections due to increased compliance by both the veterinarian and the dog owner. Strengthening cross-disciplinary communication and involvement, especially between medical and veterinary professionals, would strengthen the control of zoonotic CVBP. Holistically, these practices would contribute towards the ensuring good health and well-being of both animals and humans on the island.

## Supporting information

**S1 File. Questionnaire administered to veterinarians to obtain data on knowledge, attitude, and practices on CVBP.**
(PDF)

**S2 File. Data obtained from the knowledge, attitudes, and practices questionnaire on CVBP for Sri Lankan veterinarians.**
(XLSX)

**S1 Table. Responses provided by veterinarians in Sri Lanka (n = 170) to knowledge evaluation statements regarding canine vector-borne pathogens.**
(PDF)

**S2 Table. Vector-borne pathogens endemic to Sri Lanka according to the scientific literature up until December 2021, and a summary of responses obtained from Sri Lankan veterinarians through a knowledge, attitude, and practices survey around canine vector-borne pathogens.**
(PDF)

**S3 Table. Responses of canine practitioners in Sri Lanka on diagnosis, treatment, control, and the prognosis of CVBP infections in dogs.**
(PDF)

**S4 Table. Demographic factors, confidence in profession and self-rated knowledge associated with knowledge around canine vector-borne pathogen infections determined through univariable ordinal regression based on responses of veterinarians in Sri Lanka for a knowledge, attitude, and practices survey.**
(PDF)

**S5 Table. Associations of knowledge, practice type, confidence in profession, experience, canine caseload, and monetary constraints with practices on diagnosis, treatment, and control of canine vector-borne pathogen infections determined through univariable logistic regression based on responses of canine practitioners in Sri Lanka for a knowledge, attitude, and practices survey.**
(PDF)

## Acknowledgments

The authors would like to thank all those who participated in the pilot and the survey.

## Author Contributions

**Conceptualization:** Ushani Atapattu, Rebecca J. Traub, Anke Wiethoelter.

**Data curation:** Ushani Atapattu.

**Formal analysis:** Ushani Atapattu, Anke Wiethoelter.

**Investigation:** Ushani Atapattu, Anke Wiethoelter.

**Methodology:** Ushani Atapattu, Anke Wiethoelter.

**Project administration:** Anke Wiethoelter.

**Resources:** Anke Wiethoelter.

**Supervision:** Vito Colella, Rebecca J. Traub, Anke Wiethoelter.

**Validation:** Ushani Atapattu, Anke Wiethoelter.

**Visualization:** Ushani Atapattu, Anke Wiethoelter.

**Writing – original draft:** Ushani Atapattu.

**Writing – review & editing:** Vito Colella, Anke Wiethoelter.

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
