## [Decision Letter · Decision Letter 0]

11 Jun 2024

Dear Ms Atapattu,

Thank you very much for submitting your manuscript "Knowledge, attitude and practices of veterinarians towards canine vector-borne pathogens in Sri Lanka" for consideration at PLOS Neglected Tropical Diseases. As with all papers reviewed by the journal, your manuscript was reviewed by members of the editorial board and by several independent reviewers. The reviewers appreciated the attention to an important topic. Based on the reviews, we are likely to accept this manuscript for publication, providing that you modify the manuscript according to the review recommendations. 

The authors have clearly presented the study objectives, methods and process in a way that could be reproduced, which may be helpful in translating this work to other regions impacted by transmission of canine vector-borne pathogens. Please address the reviewer's feedback and provide additional quantitative results supporting the statement (L254-256) around the process of multivariable model selection, specifically identifying the best multivariable model to explain the data, such as how many terms were in the multivariable model(s) prior to backwards-stepwise elimination of variables. Please also clarify the description of potential confounding variables and lightly describe why a mixed model regression (e.g. model includes random effects as well as fixed effects) was not used to address anticipated covariance patterns in the data beyond the central variables of interest.

Sincerely,

Marco Coral-Almeida, M.Sc., Ph.D.

Academic Editor

Amy Gilbert

Section Editor

Reviewer's Responses to Questions

**Key Review Criteria Required for Acceptance?**

**Methods**

-Are the objectives of the study clearly articulated with a clear testable hypothesis stated?

-Is the study design appropriate to address the stated objectives?

-Is the population clearly described and appropriate for the hypothesis being tested?

-Is the sample size sufficient to ensure adequate power to address the hypothesis being tested?

-Were correct statistical analysis used to support conclusions?

-Are there concerns about ethical or regulatory requirements being met?

Reviewer #1: Yes for all the questions

Reviewer #2: Please see Summary and General Comments

Reviewer #3: Objectives, study design, and population are clearly described and appropriate. The sample size and statistical analysis are adequate to support the hypothesis being tested and the conclusions.

No ethical or regulatory concerns are present.

Just a comment on the period of collection of survey responses (from April to November 2021): overall, 170 completed surveys were collected and included for analysis (8.3% of veterinarians registered to practice in Sri Lanka). Could this data have been influenced by the pandemic conditions related to SARS-CoV-2 in Sri Lanka in 2021? I recommend discussing in the paper the potential reasons for the moderate adherence of veterinarians to the questionnaire. 

However, the study represents an excellent framework of the knowledge on CVBPs by veterinary practiotioners in Sri Lanka.

**Results**

-Does the analysis presented match the analysis plan?

-Are the results clearly and completely presented?

-Are the figures (Tables, Images) of sufficient quality for clarity?

Reviewer #1: Yes for all the questions

Reviewer #2: Please see Summary and General Comments

Reviewer #3: Results are clearly and adequately presented; tables and images are useful

**Conclusions**

-Are the conclusions supported by the data presented?

-Are the limitations of analysis clearly described?

-Do the authors discuss how these data can be helpful to advance our understanding of the topic under study?

-Is public health relevance addressed?

Reviewer #1: Yes for all the questions

Reviewer #2: Please see Summary and General Comments

Reviewer #3: Conclusions are supported by results and Authors discussed adequately their results: this will be useful for implemented continuing veterinary education, considering the public health relevances of CVBP.

**Editorial and Data Presentation Modifications?**

Reviewer #1: Abstract

Please change "Canine vector-borne pathogens (CVBP) are highly prevalent " in (CVBP) have a worldwide distribution with higher prevalence ..

Line 18 Please change as above

Line 59 The role of the dog as a resrvoir for R.conorii is still controversial. Riphicephalus sanguineosu is probably the real reservoir and the dog probably has only a marginal role. Please rephrase this as "Dogs have or may have a role as a reservoir"

Reviewer #2: Please see Summary and General Comments

Reviewer #3: Line 64: Babesia vogeli

Line 180: Trypanosoma evansi

The acronym KAP is not use at line 281

**Summary and General Comments**

Reviewer #1: Very interesting paper that explores the role of the veterinarians and pet care in the control and prevention of vector-borne zoonotic diseases from a "One Health" perspective and how the level of knowledge can depend on various factors.

It may be a starting point for future investigations that compare the levels of knowledge on the same topics by Physicians

Reviewer #2: Title – replace practice with practices – change accordingly throughout the manuscript

Keywords – display alphabetically

Line 21 – adapt to read as: making these pathogens emerging public health ISSUES

Line 28 – which species of Brugia

Line 44 – delete: “by bugs like”

Line 48 – replace controlling with preventing

Line 52 – adapt: diagnosis, treatment, and prevention of

Line 58 – Babesia spp. (instead of Babesia species)

Line 62 – adapt: Dirofilaria, WHICH infect

Line 64 – write out in full every species name at its first use, e.g. Babesia vogeli

Line 72 – delete control or delete treatment

Line 103 – these seven volunteers were not included in the final results, right?

Line 119 – 11 knowledge questions

Line 162 – replace rate with proportion

Line 162 – 8.3% is a rather low proportion – please address this issue

Line 183 – replace However with Nevertheless

Line 194 – WERE important

Line 288 – define Illinois and Ohio as USA states

Liner 384 – Conclusions

Lines 604 and 606 – is there any difference between S1 File and S1 Table?

Reviewer #3: The paper “Knowledge, attitude and practices of veterinarians towards canine vector-borne

pathogens in Sri Lanka” describes how veterinarians perceive and address CVBPs in clinical practice. Thought a questionnaire survey, the Authors assessed the KAP of veterinary practitioners and promoted a continuing veterinary education. The scientific soundness is good. The limitations of the study, well described in the text, are correlated to the period of data collection and the unpredictable response rate to the questionnaire.

PLOS authors have the option to publish the peer review history of their article (what does this mean?). If published, this will include your full peer review and any attached files.

Reviewer #1: Yes: Luigi Venco DVM, EVPC Diplomate, , EBVS® - European Veterinary Specialist in Parasitology

Reviewer #2: No

Reviewer #3: No

Figure Files:

Data Requirements:

Reproducibility:

References

---

## [Decision Letter · Decision Letter 1]

12 Jul 2024

Dear Ms Atapattu,

We are pleased to inform you that your manuscript 'Knowledge, attitude, and practices of veterinarians towards canine vector-borne pathogens in Sri Lanka' has been provisionally accepted for publication in PLOS Neglected Tropical Diseases.

Best regards,

Marco Coral-Almeida, M.Sc., Ph.D.

Academic Editor

Amy Gilbert

Section Editor

Reviewer's Responses to Questions

**Key Review Criteria Required for Acceptance?**

**Methods**

-Are the objectives of the study clearly articulated with a clear testable hypothesis stated?

-Is the study design appropriate to address the stated objectives?

-Is the population clearly described and appropriate for the hypothesis being tested?

-Is the sample size sufficient to ensure adequate power to address the hypothesis being tested?

-Were correct statistical analysis used to support conclusions?

-Are there concerns about ethical or regulatory requirements being met?

Reviewer #1: Yes for al the questios

Reviewer #2: OK.

**Results**

-Does the analysis presented match the analysis plan?

-Are the results clearly and completely presented?

-Are the figures (Tables, Images) of sufficient quality for clarity?

Reviewer #1: Yes for al the questions

Reviewer #2: OK.

**Conclusions**

-Are the conclusions supported by the data presented?

-Are the limitations of analysis clearly described?

-Do the authors discuss how these data can be helpful to advance our understanding of the topic under study?

-Is public health relevance addressed?

Reviewer #1: Yes for al the questione

Reviewer #2: OK.

**Editorial and Data Presentation Modifications?**

Reviewer #1: (No Response)

Reviewer #2: OK.

**Summary and General Comments**

Reviewer #1: Now worthy of publication in my opinion

Reviewer #2: OK.

PLOS authors have the option to publish the peer review history of their article (what does this mean?). If published, this will include your full peer review and any attached files.

Reviewer #1: **Yes: **Luigi Venco

Reviewer #2: No

---

## [Editor Report · Acceptance letter]

24 Jul 2024

Dear Ms Atapattu,

We are delighted to inform you that your manuscript, "Knowledge, attitude, and practices of veterinarians towards canine vector-borne pathogens in Sri Lanka," has been formally accepted for publication in PLOS Neglected Tropical Diseases.

Best regards,

Shaden Kamhawi

co-Editor-in-Chief

Paul Brindley

co-Editor-in-Chief
